# Applying Compost Biochar for Gas Adsorption—Effects of Pyrolysis Conditions

**DOI:** 10.3390/molecules30163365

**Published:** 2025-08-13

**Authors:** Sylwia Stegenta-Dąbrowska, Marta Galik, Magdalena Bednik-Dudek, Ewa Syguła, Katarzyna Ewa Kosiorowska

**Affiliations:** 1Department of Applied Bioeconomy, Wrocław University of Environmental and Life Sciences, Chełmońskiego Street 37a, 51-630 Wrocław, Poland; sylwia.stegenta-dabrowska@upwr.edu.pl (S.S.-D.); 120419@student.upwr.edu.pl (M.G.); ewa.sygula@upwr.edu.pl (E.S.); 2Institute of Soil Science, Plant Nutrition and Environmental Protection, Wrocław University of Environmental and Life Sciences, Grunwaldzka Street 53, 50-375 Wrocław, Poland; magdalena.bednik@upwr.edu.pl

**Keywords:** biochar, carbon monoxide, compost utilization, compost valorization, gas adsorption, gas purification, greenhouse gas, pyrolysis

## Abstract

Not all produced compost meets established quality standards, often resulting in environmental challenges. This study investigated the potential of using mature compost as a feedstock for biochar production, with a focus on evaluating the gas adsorption properties of the resulting biochars. Mature compost was utilized as a substrate, and the pyrolysis process involved heating samples within a temperature range of 400–650 °C, at 50 °C intervals, with heating rates of 10 °C·min^−1^, 15 °C·min^−1^, or 20 °C·min^−1^ for a duration of 60 min. The resulting biochars were tested for their adsorption performance against a synthetic gas mixture simulating composting emissions (CO_2_, CO, H_2_S, NH_3_, CH_4_ in N_2_). Our findings reveal a significant correlation between the pyrolysis temperature and the sorption characteristics of compost biochars. Specifically, biochars produced at temperatures of 550 °C, 600 °C, and 650 °C (with a heating rate of 10 °C·min^−1^) demonstrated the highest efficacy in reducing emissions of CO_2_, CH_4_, and H_2_S, achieving reductions of 69%, 69%, and 72%, respectively. However, these biochars exhibited lower adsorption capacity for CO and NH_3_. Interestingly, biochars produced at 400 °C and 450 °C showed enhanced performance for CO adsorption. Compost biochar shows strong potential for gas adsorption, particularly for CO, CO_2_, and H_2_S. Due to its pronounced CH_4_ sorption capacity, such biochar is better suited for mitigating emissions during composting rather than for biogas purification.

## 1. Introduction

The capture and separation process of CO_2_, H_2_S, and CH_4_ from various waste treatment sources through adsorption is widely utilized [1]. Biochar, a product of thermochemical conversion of biomass, has shown significant adsorption capacities for CO_2_, H_2_S, and CH_4_ [2]. Additionally, among the various known solid sorbents, it represents a favorable solution, both economically and environmentally. Its advantages include its low cost and easy access to feedstocks, as well as its potentially high adsorption capacity [3]. The adsorption performance of biochar is influenced by both its physical and chemical characteristics, including its surface area and functional groups [4], as well as technological parameters such as the temperature, heating rate, residence time, pressure or chemical modification method, and feedstocks used in the process [3]. During pyrolysis, components of biomass such as lignin, hemicellulose, and cellulose undergo complete or partial decomposition at various temperature ranges, contributing to the formation of the biochar structure and enhancing its aromaticity and hydrophobicity. This process also leads to the development of numerous pores of varying sizes on its surface [5,6].

Feedstocks for biochar production come from waste, biomass, or materials designated for disposal. The most common are municipal sewage sludge, the organic fraction of municipal waste, agricultural and industrial residues, animal manure, and algae [7,8]. These materials vary considerably in their composition, particularly in terms of the organic matter, nutrient content, and presence of contaminants, which in turn influence the physicochemical properties of the resulting biochar [9]. Compared to these conventional feedstocks, mature compost, especially that derived from unsold or non-compliant fractions, offers a relatively stable and homogenous matrix, with lower moisture content and volatile organic compound loads [10].

The latest material tested for biochar production is mature compost, demonstrating promising potential for reducing greenhouse gases (GHGs) during composting [11], as well as reducing volatile organic compounds (VOCs) during food waste storage [10]. The mature compost has a lower organic content compared to biochar derived from biomass, and is rich in carbon content and nutrient levels, making it ideal for use as an organic fertilizer or as an amendment to soils [12]. Additionally, compost is often generated in large quantities, and in cases where it does not meet quality standards for agricultural use [13], converting it into biochar provides an effective valorization pathway. Not all produced compost meets the required standards, potentially leading to environmental problems. However, repurposing low-quality compost that is unsuitable for agricultural use as a feedstock for biochar production offers a viable alternative to landfill disposal. This conversion not only enhances the value of the compost but also mitigates landfill waste and supports circular economy principles.

The existing body of research on biochar and greenhouse gas (GHG) adsorption has primarily focused on pure gases such as CO_2_ and H_2_S, or simple binary mixtures such as CO_2_/N_2_ [14]. However, gas emissions from waste treatment processes—such as composting and biogas production—typically consist of complex mixtures of various gases [15], not isolated components. Despite this, only a limited number of studies have investigated the behavior of biochar under such multicomponent gas conditions that more accurately reflect real-world scenarios [16]. This represents a significant gap in the current state of knowledge, as the presence of multiple gases may influence the adsorption capacity and selectivity of biochar through competitive or synergistic interactions. Moreover, existing studies tend to focus on biochar derived from biomass feedstocks [17], which differ substantially in their composition and structure from mineral-rich compost-based biochar. Addressing these research gaps is essential for improving our understanding of biochar’s performance under practical conditions and for advancing its application in environmental gas mitigation strategies. Furthermore, developing a cost-effective biochar from feedstocks such as compost addresses both the challenges of waste management and the growing demand for sustainable gas cleaning technologies. This dual benefit makes biocoal from compost particularly attractive for industrial-scale applications, especially in facilities where both composting and gas treatment are required.

The purpose of this study is to showcase the efficacy of compost biochar in adsorbing a mixture of gases released during biomass decomposition, including H_2_S, CO, CO_2_, NH_3_, and CH_4_. This work evaluates the potential of compost biochars derived with different temperatures and heating rates for pyrolysis and their effect on the gas adsorption capacity. To showcase the economic viability of the proposed comprehensive solution, we also provide an analysis of the mass and energy balance of the compost biochar.

## 2. Results and Discussion

### 2.1. Biochar Properties

In the conducted experiment, the compost biochars and compost were analyzed for their basic physicochemical properties, the results of which are presented in Table 1 and Appendix A. Although these properties mostly do not have a direct impact on the biochar’s adsorption ability, their assessment is crucial in the discussion on the impact of the substrate and pyrolysis conditions on the final product, especially in the context of possible environmental risk.

The primary raw material had a pH level close to neutral (pH 7.3), which is typical for stabilized compost [18]. In contrast, the biochars exhibited alkaline reaction, with pH values ranging from 8.4 to 9.6. It was also observed that an increase in pyrolysis temperature causes a gradual rise in the pH of biochars, as confirmed by research conducted by Wang et al. in 2015 [19]. This phenomenon is associated with increases in the content of carbonates and basic inorganic cations, as well as the release of alkaline salts and an increased ash content. This process is also related to the reduction of acidic functional groups and the emergence of basic functional groups, which further enhances the alkalinity of the biochars [20].

Analyzing the ash content, it can be observed that it increases in biochars as the pyrolysis temperature rises, compared to the primary raw material. The highest ash content, reaching 90.8%, was recorded for the biochar produced at a temperature of 550 °C and a heating rate of 15 °C·min^−1^. In the biochar obtained at 650 °C, the ash content was only slightly lower. Research by Zielińska et al. in 2015 [21] confirmed a similar trend; in biochars from sewage sludge, the ash content also increased compared to the primary raw materials. In general, the ash content in compost-derived biochars is relatively high compared with other green feedstock, where typical values do not exceed 10–30% [22]. The compost itself had a high content of mineral fractions, which then had impact on the other properties of the resulting biochar. This is also reflected in the carbon content, which originates from the organic fraction of the raw material. Typical compost contains 13% C, whereas in biochars the values range between 8 and 15%. According to the criteria of the International Biochar Initiative (IBI), obtained biochars mostly contain a minimum carbon content of 10% [23]. However, no clear trend was observed between the pyrolysis temperature and the carbon content. Although many studies report that the carbon content increases with rising temperature due to the intensification of the carbonization process [24], for biochars produced from materials with low organic matter content, even a decrease in carbon content is characteristic. This is confirmed by research conducted by Zielińska et al. (2015) on biochars from sewage sludge and by Cantrell et al. (2012) on biochars from poultry manure [21,25]. This behavior is attributed to the low organic matter and high mineral (ash) contents of the compost. During pyrolysis, a significant portion of the limited organic carbon is volatilized as gases, while the mineral fraction remains, increasing the relative ash content. As a result, the proportion of carbon in the final biochar does not increase with temperature and may even decrease. Similar trends have been reported for other high-ash feedstocks such as sewage sludge and poultry manure [25].

In comparison to compost, which contained macronutrients in various amounts for Ca (17,760 mg·kg^−1^ d.m.), K (6155 mg·kg^−1^ d.m.), Mg (2697 mg·kg^−1^ d.m.), Na (1793 mg·kg^−1^ d.m.), and P (5835 mg·kg^−1^), most of the analyzed biochars exhibited increases in their contents. The highest concentrations of these elements were observed in biochars produced at temperatures of 550 °C and higher, which was particularly clear for K, Na, and P. This phenomenon can be attributed to the thermal transformation of the raw material [10]. The pyrolysis of compost samples leads to the densification of the substances contained within them, which in turn contributes to the increased amounts of the analyzed elements. Moreover, as noted by Tomczyk et al. 2020 [26], the increasing trend of certain mineral components in biochars, such as calcium, potassium, sodium, and magnesium, is related to the rise in pyrolysis temperature. However, the authors emphasize the importance of the raw materials used for biochar production, indicating a correlation between the increased contents of inorganic substances and ash and the rise of these elements. In the analyzed case, when the compost contained only 24% of organic components (based on LOI), with the dominant mineral fraction, the densification effect in the pyrolysis process was not clear. Moreover, no correlation was observed between the variable heating rate and macronutrients concentration.

#### 2.1.1. BET and Sorption Isotherms

A total of 18 biochar variants were characterized, revealing a clear trend; as the pyrolysis temperature increased, the specific surface area (SSA) of the biochars also increased, consistent with findings from other studies on biochar production. As shown in Figure 1, biochars produced at higher temperatures, such as 650 °C, exhibited significantly larger surface areas compared to those produced at lower temperatures, such as 400 °C. The highest surface area of 39.2 m^2^·g^−1^ was observed for biochar produced at 650 °C with a heating rate of 15 °C min^−1^. While this value is modest compared to conventional activated carbons (typically 500–1500 m^2^/g), it represents a reasonable performance for a sustainable, waste-derived material. This result aligns with those reported by Gai et al. 2014 [27], who noted that biochar surface areas often increase exponentially with the pyrolysis temperature due to the formation of more microporous structures. Conversely, the lowest surface area, 2.34 m^2^·g^−1^, was observed for biochar produced at 400 °C with a faster heating rate of 20 °C min^−1^, indicating that lower temperatures and faster heating rates inhibit the development of porosity, a phenomenon also highlighted in their comparative analysis of slow- and fast-pyrolyzed biochars. Although the surface area increased with the temperature, beyond a certain point, rapid heating rates led to pore collapse or clogging, reducing the overall surface area. The pore volume followed a similar decreasing trend with faster heating rates, likely due to insufficient time for proper pore development [28].

The obtained BET adsorption–desorption isotherms reveal that the generated biochars exhibited an increased capacity to adsorb nitrogen with rising relative pressure and pyrolysis temperature (Appendix A). Biochars produced at 400 °C demonstrated the lowest nitrogen adsorption, while those produced at 650 °C exhibited the highest. Our analysis of Appendix A indicates an initial modest increase in the nitrogen adsorption curves, followed by a gradual and sustained rise from approximately 0.2 p/p0, reaching a peak value. These findings align with type II isotherms as per the IUPAC classification, suggesting a potential microporous structure of the biochars [1]. The observed hysteresis loop type is H4, often found in mesoporous zeolites and micro-mesoporous carbons [1].

#### 2.1.2. FTIR-ATR Spectroscopy

The results show that subjecting materials to thermal treatments at different temperatures causes noticeable changes in organic matter, particularly in producing biochar. To further investigate these molecular alterations, FTIR-ATR spectra were analyzed (Figure 2). The spectra illustrated characteristic differences between the initial compost sample and compost biochars produced at various temperatures (400, 450, 500, 550, 600s and 650 °C) and heating rates (10 °C∙min^−1^, 15 °C∙min^−1^, and 20 °C∙min^−1^).

The typical FTIR spectrum for mature compost should exhibit a wide band of C–H and O–H bands present in alcohol, phenol, or carboxyl groups in the region of 2800–3600 cm^−1^, along with two characteristic bands at 2925 and 2845 cm^−1^ that may be attributed to the asymmetric and symmetric vibrations of C–H stretching of CH_3_ and CH_2_ groups [29]. The peaks observed at 1700 and 1600 cm^−1^ correspond to the stretching vibrations of C=O in carboxyl groups and aromatic C=C bonds, respectively [30]. Furthermore, the peak at 1270 cm^−1^ is indicative of phenolic–OH stretching [31]. The mentioned bands were only visible in raw compost at 400 °C; however, with pyrolysis, this group decreased and was lost, which is consistent with previous studies on biomass pyrolysis. The observed reduction indicates the degradation of lipids and carbohydrates, resulting in an increase in the presence of aromatic groups—strong absorption peaks at 750–870 cm^−1^, attributed to stretching vibrations of the aromatic ring of C–H or C–N, R–O–C, or R–O–CH_3_ groups [31]—were visible in all pyrolyzed samples. Due to the specific nature of the material used in biochar production, namely compost rich in mineral fractions, no distinctive characteristics typical of the pyrolysis group were observed.

At 400 °C, both compost and biochar samples exhibited a distinct sharp band at 875 cm^−1^, along with a broader band at 1400 cm^−1^, which can be attributed to CaCO_3_ [32]. However, at temperatures exceeding 450 °C, the peak at 1400 cm^−1^ disappeared, while the levels of Ca and pH increased (Table 1). This change may be attributed to the partial conversion of organic matter into gaseous compounds during pyrolysis, which results in an elevated concentration of alkali elements in a reduced mass of the produced biochar.

In contrast, a strong peak at around 1020 cm^−1^ indicates the combination of C–O stretching of polysaccharides and Si–O–Si bonds of silica and the Si–O–C group [33], which were visible in all samples. The consistent structure across all samples confirms the presence of mineral fractions in the biochars, constituting 85–90% of the ash (AC), as indicated in Table 1. The absorbance peaks generally increased with the temperature of pyrolysis across all heating times (Figure 2b,d,e).

### 2.2. Energy Balance of Compost Biochar

The mass and energy balance of the selected test samples is shown in Table 2. According to the simulation, producing 1 g of biochar at 550 °C requires 1.197 g of material, at 600 °C it requires 1.207 g, and at 650 °C it requires 1.211 g. The external energy input required for 1 g of biochar production was 287 J for biochar derived at 550 °C, 296 J for biochar at 600 °C, and 411 J for biochar at 650 °C (based on DSC; Appendix A). The energy balance indicates a reduction in the calorific value of the products from the pyrolysis process (Table 2).

This can be attributed to the nature of the sample and the fact that it was subjected to the process twice. Rafiq et al., in 2016, noted the energy content of the raw material used for biochar production exceeds the energy in the final product [33]. This finding is consistent with our observations, where biochar produced at higher temperatures exhibited reduced combustible properties due to increased gas production, which simultaneously required more external energy. However, the gas generated during the process contains significant energy, which can be reused in the system. In this study, biochar produced at 650 °C with a heating rate of 10 °C∙min^−1^ required 411 J of external energy, as compared to 287 J at 550 °C. This energy balance assessment helps determine the efficiency of the process by analyzing the substrate requirements for producing a unit of biochar and the energy required for its production. For instance, producing 1 ton of biochar from compost at 550 °C and a heating rate of 10 °C min^−1^ would require 1.197 tons of compost and 287 MJ of energy. The energy contained in process gases could be utilized to enhance the overall system efficiency, such as for substrate drying [34].

### 2.3. Sorption Tests

Due to the growing demand for specialized sorption materials that can be used in environmental applications, the compost biochar was evaluated as a potential adsorbent for gas treatment. The sorption tests were conducted to investigate the gas adsorption capacity of the biochars produced under various pyrolysis conditions. Such tests targeted the simultaneous removal of multiple gases commonly associated with biomass decomposition, including CO_2_, H_2_S, CO, NH_3_, and CH_4_.The breakthrough curves for each gas and biochar type provided insight into the adsorption efficiency and effectiveness of the different variants of biochar.

#### 2.3.1. CO_2_

The capacity of biochar to capture and retain CO_2_ through sorption processes plays a crucial role in evaluating its efficacy as a carbon sequestration material (Appendix A). In the present study, the highest degree of CO_2_ reduction of 69% was observed for a variant of biochar produced at 650 °C with a temperature increase rate of 10 °C∙min^−1^ (Appendix A). BC650/20 reduced the CO_2_ emissions by 55%, while BC650/15 showed stable adsorption, with slight variations between 60 and 420 s of 49%. The worst CO_2_ adsorption was observed for the variants generated at the lowest temperature (<450 °C) (Appendix A). Their adsorption capacities initially ranged from 8% for BC400/10 to 30% for BC400/20. It was also observed that the amount of CO_2_ adsorbed by the biochars produced with a heating temperature rate of 10 °C∙min^−1^ grew with the increasing temperature of the pyrolysis process. The Tukey test for the average CO_2_ sorption values after 10 min of testing, showed that there were statistically significant differences (*p* < 0.05) between the groups (Figure 3a). Notably, the capacity for CO_2_ adsorption increases with increasing pyrolysis temperature, but only up to a temperature of 550 °C. Further increases in temperature are no longer effective and do not improve these properties.

#### 2.3.2. CO

The study of CO adsorption by compost biochars revealed several significant findings that are crucial for understanding the efficacy of these materials in gas purification applications (Appendix A). Most adsorption curves showed an initial decrease followed by a slight increase towards the end of the measurement period, with biochars BC450/10 and BC600/10 exhibiting the most pronounced upward trend (from 36% to 44% and 29% to 34%, respectively). Among all tested biochars, BC400/10, BC450/10, and BC400/20 demonstrated the highest CO reduction results, with adsorption rates of 46–50%, 36–44%, and 37–38%, respectively. Conversely, biochars BC400/15, BC450/15, and BC650/15 showed the poorest adsorption performance, with initial and final CO adsorption value ranges of 16–17%, 20–17%, and 23–21%, respectively. A notable trend was observed in the relationship between the heating rate and adsorption efficiency. Biochars produced with a heating rate of 10 °C∙min^−1^ generally exhibited superior CO adsorption compared to those produced at 15 °C∙min^−1^ and 20 °C∙min^−1^, with the exception of BC550/10. This trend was consistent across most pyrolysis temperatures, including 400 °C, 450 °C, 600 °C, and 650 °C.

Tukey’s test revealed significant differences (*p* < 0.05) in CO sorption among biochar variants after 10 min of testing (Figure 3b). Notably, the CO sorption results for BC400/15 and BC450/15 differed significantly. These findings indicate that the 400/10 variant was statistically the most effective in CO adsorption. Importantly, the study revealed an inverse relationship between the pyrolysis temperature and CO adsorption efficiency, with the adsorption capacity significantly decreasing as the pyrolysis temperature increased. This trend suggests that lower pyrolysis temperatures, particularly around 400 °C, may be optimal for producing biochars with enhanced CO adsorption properties.

#### 2.3.3. H_2_S

Biochars produced at higher temperatures (550 °C, 600 °C, and 650 °C) demonstrated superior H_2_S adsorption compared to those produced at lower temperatures (400 °C and 450 °C) (Appendix A). BC650/10 exhibited the highest H_2_S reduction, with 74% adsorption, while BC500/15 exhibited the lowest H_2_S reduction overall at 22%. Tukey’s test revealed significant differences (*p* < 0.05) in H_2_S sorption among biochar variants after 10 min, particularly between BC400/15, BC450/15, BC450/20, and BC500/10, and BC550/20 and BC600/10, as shown in Figure 3. Notably, the H_2_S sorption capacity increased with the pyrolysis temperature up to 550 °C, beyond which further temperature increases did not significantly improve the adsorption properties, except for BC650/10. These findings suggest that moderate to high pyrolysis temperatures (550–650 °C) with varying heating rates produce biochars with superior H_2_S adsorption properties, with an apparent optimum around 550 °C.

#### 2.3.4. NH_3_

The sorption curves (Appendix A) showed a decreasing trend over time, indicating increasing NH_3_ emissions and diminishing adsorption capacities. Biochars BC400, BC450, BC600, and BC650 with a 10 °C∙min^−1^ heating rate demonstrated the highest NH_3_ emission reduction, while BC500 and BC550 performed best with a 15 °C∙min^−1^ heating rate. The most effective was BC450/10, initially adsorbing 87% of NH_3_, although this decreased to 56% by the end of the measurement. Conversely, BC550/20 initially showed the lowest adsorption rate (22%), while BC400/15 had the lowest final value of 1%. Generally, biochars produced with a 20 °C∙min^−1^, exhibited the poorest NH_3_ adsorption. Notably, lower pyrolysis temperatures, especially around 400 °C, resulted in superior NH_3_ adsorption efficiency (Figure 3d). These findings underscore the complex relationship between biochar production parameters and NH_3_ adsorption performance, highlighting the importance of lower pyrolysis temperatures and slower heating rates for optimizing NH_3_ removal capabilities, which may be related to the retention of more functional groups. At lower temperatures, fewer organic functional groups are destroyed, leaving more sites for NH_3_ to interact with. Another possible reason is incomplete carbonization, which results in a more diverse and reactive surface chemistry [35].

#### 2.3.5. CH_4_

The results demonstrate a statistically significant affinity for CH_4_ adsorption (*p* < 0.05), as illustrated in Figure 3e and Appendix A. Notably, these variations did not exhibit a positive correlation with increasing pyrolysis temperature. Remarkably, the highest adsorption capacity was attained at lower temperature ranges during biochar production. Among the variants tested, BC400/10, BC400/15, and BC400/20 achieved CH_4_ adsorption capacity rates of 65%, 67%, and 58%, respectively. In contrast, the BC500/15 variant displayed the lowest CH_4_ adsorption capacity, reaching only 20% (refer to Appendix A).

A specific trend regarding CH_4_ adsorption was observed, indicating an improvement in sorption properties over time. For instance, the samples BC650/10, BC650/15, and BC650/20 initially adsorbed 59%, 38%, and 30% of CH_4_, respectively; these values increased to 63%, 40%, and 45% after a duration of 10 min.

### 2.4. Sorption Mechanism

The CO_2_ adsorption capacity of biochar is strongly influenced by its physical properties and chemical characteristics, which are determined by the feedstock and pyrolysis conditions. The pyrolysis temperature has a significant effect on CO_2_ adsorption, with optimal performance typically achieved between 500 and 700 °C [36]. The enhanced adsorption at moderate temperatures (550–650 °C) can be attributed to several factors. Despite the modest surface area values compared to commercial activated carbons, the increases in surface area and microporosity volume due to volatile release and pore formation provide cost-effective adsorption performance [37]. Secondly, higher pyrolysis temperatures promote the formation of aromatic carbon structures that can enhance CO_2_ adsorption through π–π interactions [38]. Additionally, at moderate pyrolysis temperatures, some oxygen-containing functional groups are removed from the surface of the biochar, which can reduce its polarity and make it more favorable for CO_2_ adsorption through non-polar interactions [39].

However, it is worth noting that some studies suggest that the presence of oxygen-containing functional groups, particularly carboxyl and hydroxyl (Figure 3), can actually promote CO_2_ adsorption by increasing the polarity of the adsorbent [37]. The study also showed that slower heating rates generally produce biochar with better CO_2_ adsorption compared to faster heating rates. Slower heating rates allow for better development of the pore structure and the preservation of favorable surface chemical properties [37].

The adsorption of CO_2_ on biochar involves both physisorption, based on weak van der Waals forces, and chemisorption, involving stronger chemical bonds. The presence of metal oxides, such as MgO (Table 1), can play an important role in CO_2_ adsorption by chemically reacting with CO_2_ to form carbonate compounds, significantly increasing the adsorption capacity of the biochar [17].

Regarding H_2_S adsorption, the study revealed a noticeable tendency for the adsorption capacity to increase with higher pyrolysis temperatures, particularly in the 550–650 °C range. This observation suggests that the structural and chemical changes occurring at these elevated temperatures are beneficial for H_2_S adsorption, likely due to greater pore and surface development [40]. The effect of the heating rate on H_2_S adsorption was not consistent at all pyrolysis temperatures, suggesting a complex interaction between the heating rate and final temperature in determining biochar properties [41].

In contrast to CO_2_ and H_2_S, CO adsorption showed an inverse relationship with the pyrolysis temperature. Biochars produced at lower temperatures, particularly 400 °C and 450 °C, showed the highest CO adsorption capacity. This observation may be related to the behavior of functional surface groups at lower pyrolysis temperatures, particularly carboxyl groups (Figure 3), which have shown the lowest binding energy values of all the functional groups, indicating their greatest contribution to CO sorption [42].

The differences between the effects of temperature on gas adsorption are due to different mechanisms; CO_2_ and H_2_S undergo physical adsorption with increased surface areas, as well as chemisorption with basic metal oxides (CaO, MgO), which are concentrated at higher temperatures [43]. CO and NH_3_, on the other hand, require oxygen-containing functional groups (carboxyl, hydroxyl) for chemical interaction, as confirmed by the FTIR analysis showing that these groups are most abundant at 400–450 °C and decompose at higher temperatures [44]. This explains why the adsorption of CO and NH_3_ decreases with temperature, whereas the adsorption of CO_2_ and H_2_S increases, demonstrating a critical interaction between thermal treatment, surface chemistry, and gas-specific sorption mechanisms.

The chemical and physical changes that occur in the material during pyrolysis are related to the processes of dehydration and devolatilization. At lower temperatures, these processes are less complete, affecting the preservation of more oxygen-containing functional groups, which are crucial in the NH_3_ sorption process [35]. Lower pyrolysis temperatures result in reduced aromatic structures, preserving more aliphatic groups that can interact with NH_3_ [45]. Lower temperatures may create a pore structure more suitable for NH_3_ adsorption, with a higher proportion of micropores [45].

A slower heating rate (10 °C∙min^−1^) provides better performance for NH_3_ adsorption. This may be related to the gradual decomposition of matter in such conditions, providing more uniform carbonization of biomass particles and allowing a controlled release of volatiles, which ultimately results in the formation of a more homogeneous pore structure [45]. Slower heating may also allow the preservation of functional groups that interact with NH_3_ [45].

The low absorption efficiency of the biochar produced at 20 °C∙min^−1^ was most likely due to the rapid release of volatiles, which can cause the structure to collapse or block pores [46]. Uneven heat distribution can in turn lead to inconsistent carbonization and a less optimal surface chemistry [47].

For CH_4_ adsorption, the results indicated the highest capacity for the biochar produced at 400 °C with a temperature heating rate of 20 °C°C∙min^−1^. This may be related to the retention of more functional groups favoring the adsorption of this gas, as well as the presence of CaCO_3_ and the correlated alkaline pH [48]. The alkaline pH resulting from the presence of CaCO_3_ may favor the deprotonation of the functional groups, increasing their negative charge and enhancing their ability to interact with positively charged molecules [49]. The stability of the CH_4_ adsorption over time varied with the pyrolysis temperature, with the biochars produced at 400 °C, 450 °C, and 600 °C showing stable adsorption, in contrast to the biochars produced at 500 °C, 550 °C, and 650 °C, which showed variability. This difference may be related to the potential effect of the pyrolysis temperature on the pore structure and biochar surface stability [50].

Detailed studies of the gas sorption of compost biochars reflect large adsorption efficacy disparities among gaseous compounds; H_2_S, CO_2_, and CH_4_ mostly show much higher sorption rates compared with CO and NH_3_. The result gives further evidence for the complexity of the interactions between properties of biochar and specific characteristics of a particular gas, pointing at the possibility to obtain biochars specifically tailored for particular environmental contaminants. The superior adsorption of H_2_S, CO_2_, and CH_4_ indicates that the physical and chemical properties of compost biochars are well-suited for capturing these gases. This finding also has important implications for applications in biogas purification, the management of landfill gases, and elsewhere where those gases are common environmental pollutants. By contrast, the very low adsorption rates detected for CO and NH_3_ would suggest that further treatment or alternative synthesis paths may be required to improve the performance of these biochars in applications of this type. Previous studies have shown that the level of gas adsorption by a biochar is significantly affected by the composition of the gas mixture. The presence of different gases in a mixture can lead to competition for adsorption sites, change the adsorption kinetics of individual components, and modify the adsorption mechanisms. As a result, the adsorption capacity of biocarbon for a given gas may be different in a mixture than for a pure gas. This is particularly important in the context of practical applications, where biochar is typically exposed to gas mixtures rather than individual components. Even though our research was focused on testing mixtures of several gases instead of selecting individual gases, our results provide information about the relative affinity of compost biochar to different gases. The improved performance for CO_2_, H_2_S, and CH_4_ compared to CO and NH_3_ suggests preferential adsorption, which is crucial for practical applications. The observed differences in adsorption efficiency (69–72% for CO_2_, H_2_S, and CH_4_ compared to lower values for CO and NH_3_) indicates that compost biochar exhibits selective behavior favoring certain target gases.

## 3. Materials and Methods

### 3.1. Materials, Biochar Production, and Determination of Material Properties

For the experiment, biochar produced from the pyrolysis process of the certified organic fertilizer “Best-Terra” from BEST-EKO Sp. z o. o. was utilized (Figure 4a). This fertilizer is made from 90% green waste and 10% sewage sludge. Initially, the compost underwent a drying process (Figure 4b). The mixture was then sieved through a 2 mm mesh. The resulting material samples were placed in heat-resistant vessels in the SNOL 8.1/1100 muffle furnace (SNOL, model 8.1/1100, Utena, Lithuania). The process of producing biochars from compost is illustrated in Figure 4a,b.

The pyrolysis process involved heating the samples within a temperature range of 400–650 °C at intervals of 50 °C, at rates of 10 °C·min^−1^, 15 °C·min^−1^, or 20 °C·min^−1^ for a duration of 60 min, with a continuous flow of CO_2_. Based on our previous research [51], pyrolysis of biodegradable waste should be carried out at 400–650 °C to optimize the biochar yield and sorption properties. A moderate heating range (10–20 °C/min) and a residence time of 60 min are recommended to maximize solid product formation [2]. The incorporation of CO_2_ in the pyrolysis process of waste and biomass not only increases the gas yield but also minimizes tar production and improves the overall product distribution [52]. After reaching room temperature, the samples were taken out of the muffle furnace and stored in airtight containers. Subsequently, both the compost and the resulting biochars underwent various physicochemical analyses, including of the moisture content (MC), volatile solids (VS), pH, elemental composition (C, H, N, S, O), and contents of micro- (Pb, Cd, Zn, Cr, Ni, Cu, Mn, and Hg) and macronutrients (P, Ca, K, Mg, and Na) and functional group (FTIR). Furthermore, the biochars were examined for SSA using the BET method, whereby sorption and desorption isotherms were determined. All employed methodologies, along with their respective parameters and equipment, are comprehensively outlined in Appendix A.

### 3.2. Thermogravimetric and Differential Scanning Calorimetry Analyses

After sample characterization, the dry compost sample was subjected to a thermogravimetric analysis (TGA) and differential scanning calorimetry (DSC) analysis. The TGA and DSC analyses were performed using a thermal gravimetric analyzer (TA Instruments, SDT Q600, New Castle, DE, USA). The dry sample of around 15 mg was placed into a corundum crucible that was next placed into the analyzer. The sample was heated up from room temperature ~20 °C to 650 °C with heating rates of 10, 15, and 20 °C·min^−1^. The argon at a flow of 100 mL·min^−1^ was used to provide inert conditions. Next, the derivative thermogravimetry (DTG) was determined using TGA data.

### 3.3. Mass and Energy Balance of the Pyrolysis Process

To assess the mass and energy balance for the pyrolysis process, the substrate was first prepared, and its properties were determined based on previous laboratory studies. The higher heating value (HHV) and mass yield (MY) of the process were measured to facilitate further calculations. Furthermore, a differential scanning calorimetry (DSC) analysis was performed to determine the amount of external energy required to heat the substrate during the pyrolysis process. The mass yield (MY) of the pyrolysis process was calculated as the ratio of the mass of biochar to the mass of the substrate. Using this value, the mass of the substrate needed to produce 1 g of biochar was determined. After determining the substrate mass, the energy contained in the substrate was calculated using its higher heating value (HHV) by multiplying it by the substrate’s mass. This energy represents the amount of chemical energy available in the substrate before the pyrolysis process. To calculate the external energy required for biochar production, a DSC analysis was used. This analysis allowed for determining the amount of energy needed to heat the substrate to the specific temperature of the pyrolysis process. The obtained external energy values were incorporated into the calculations as a key component of the energy balance. Subsequently, the energy contained in the produced biochar was calculated. The higher heating value of biochar, obtained from previous studies, was used for this purpose. The mass of the biochar was multiplied by its higher heating value, representing the amount of chemical energy stored in 1 g of the produced biochar. Next, the energy contained in the gas was calculated, considering both its chemical and thermal energy. The chemical energy of the gas was determined based on its composition and the heating value obtained from the literature data. The thermal energy was calculated by considering the specific heat of the gas and its temperature change during the process [53].

After determining all the energy components (substrate energy, external energy, biochar energy, and gas energy), they were combined in the mass and energy balance. Based on this, a table was created to present the results for various process parameters, such as pyrolysis temperature and the amount of external energy required to carry out the process.

### 3.4. Adsorption Tests

The adsorption tests were conducted using an external filter packed with compost biochar (Figure 4b). The filter was exposed to a gas mixture designed to simulate the primary components of composting gas (CH_4_, CO_2_, H_2_S, CO, NH_3_ in N_2_). The testing setup comprised four key components: a fixed-bed sorber (filled with biochar), a mass flow regulator, and a portable gas analyzer. The scale (Radwag PS 3500.R2, Poland; ±0.01 g) was used to weigh 1 g of compost biochar and place it into a tightly closed container with filters, mounted on a laboratory stand. An analyzer and a cylinder containing a gas mixture were connected to the container. Constant flow was established in the cylinder reducer, which was monitored on a rotameter. The concentrations of selected gases (CO_2_, CO, H_2_S, NH_3_, CH_4_) passing through the biochar layer were measured for 10 min, with the results recorded on the analyzer every 30 s. This process was repeated for all types of biochars in three repetitions, with each repetition involving a new sample of the tested raw material. The configuration of the gas measurement process using biochars is illustrated in Figure 4b.

During the experiment, two gas mixtures were employed. The first comprised 300 ppm mol/mol (±3 ppm mol/mol) H_2_S, 10% mol/mol (±0.02% mol/mol) CH_4_, 15% mol/mol (±0.03% mol/mol) CO_2_, and 74.97% mol/mol (±0.15% mol/mol) N_2_. The second mixture consisted of 300 ppm mol/mol (±1.5 ppm mol/mol) CO, 300 ppm mol/mol (±3 ppm mol/mol) NH_3_, and 99.94% mol/mol (±0.20% mol/mol) N_2_. Concentration measurements of selected gases (CO_2_, CO, H_2_S, NH_3_, CH_4_) were conducted using the Nanosens DP-28 BIO portable electrochemical gas analyzer. Concentrations of CO_2_, CO, H_2_S, NH_3_, and CH_4_ were determined in ppm in the following ranges: CO_2_ 0–100% (±2%), CO 0–2000 ppm (±20 ppm), H_2_S, NH_3_ 0–1000 ppm (±10 ppm), and CH_4_ 0–100% (±1%).

### 3.5. Statistical Analysis

For the statistical analysis, 13.3 Statistica software (TIBCO Software Inc., Palo Alto, CA, USA) was used. For the statistical differences of the contributions of VOCs, a one-way ANOVA was applied according to Tukey’s test at a significance level of *p* < 0.05, including previous verification of the normality and homogeneous variance using the Levene test. For all relevant cases, standard deviation (SD) was applied.

## 4. Conclusions

In this study, we investigated the potential application of compost biochar for the adsorption of CH_4_, CO_2_, NH_3_, H_2_S, and CO. Our findings demonstrate the significant influence of the pyrolysis temperature on the sorption characteristics of compost-derived biochars. Specifically, the biochars produced at temperatures of 550 °C, 600 °C, and 650 °C (with a heating rate of 10 °C min^−1^) exhibited the highest efficacy in reducing emissions of CO_2_, CH_4_, and H_2_S, achieving reductions of 69%, 69%, and 72%, respectively. In contrast, the effectiveness of these biochars for CO and NH_3_ adsorption was comparatively lower. Notably, biochars produced at 400 °C and 450 °C displayed superior adsorption properties, specifically for CO emissions. However, due to its significant capacity for CH_4_ sorption, biochar is more likely to be utilized for mitigating emissions during composting rather than for biogas purification.

This study confirms the industrial potential of compost biochar for purifying gases from biological waste treatment. Scaling up requires further optimization, including testing bed spraying or biochar activation processes. Surplus compost that fails to meet quality standards offers a low-cost filler alternative for sorption. Although the pyrolysis of compost biochar involves costs, these may be offset by savings from reduced landfill fees and odor regulation compliance. Although the surface areas achieved (up to 39.2 m^2^/g) are modest compared to commercial activated carbons, the combination of reasonable adsorption performance, sustainability benefits, and cost-effectiveness positions compost biochar as a viable alternative for environmentally conscious gas treatment applications.

## Figures and Tables

**Figure 1 molecules-30-03365-f001:**
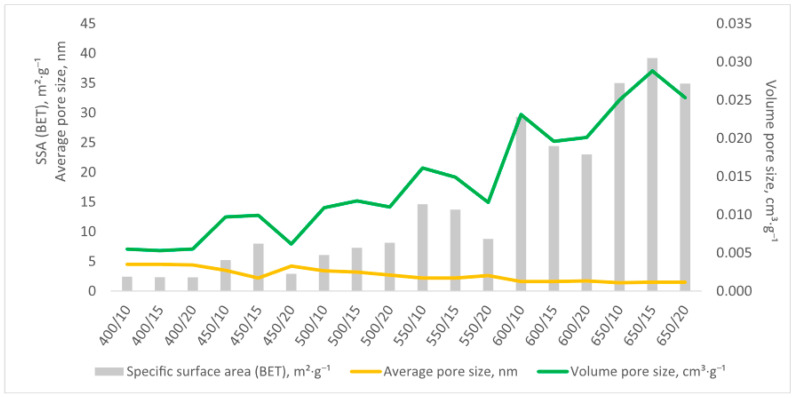
SSA (BET), volume, and average pore size values of the tested biochar variants (sorption–desorption isotherms are presented in Appendix A).

**Figure 2 molecules-30-03365-f002:**
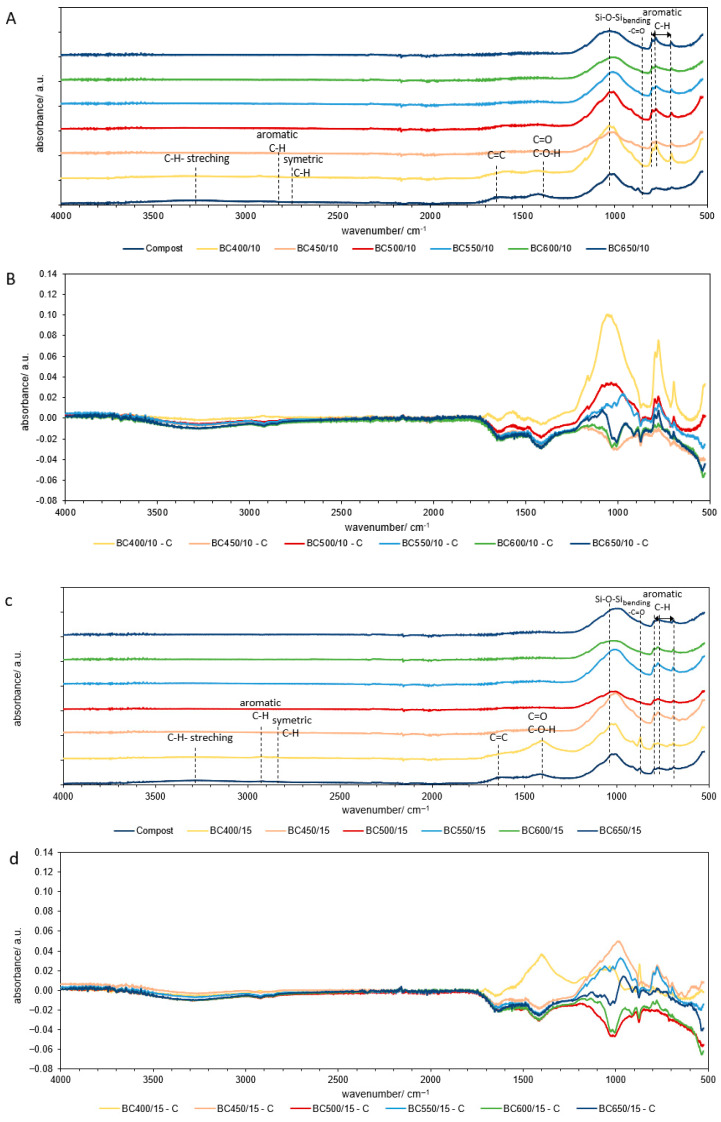
FTIR results for compost and compost biochars: (**a**) the FTIR spectra of compost and compost biochars (BC400, BC450, BC500, BC550, BC600, BC650) produced at a heating rate of 10 °C∙min^−1^; (**b**) the subtraction spectra of biochars produced at a heating rate of 10 °C∙min^−1^; (**c**) the FTIR spectra of compost and compost biochars (BC400, BC450, BC500, BC550, BC600, BC650) produced at a heating rate of 15 °C∙min^−1^; (**d**) the subtraction spectra of biochars produced at a heating rate of 15 °C∙ min^−1^; (**e**) the FTIR spectra of compost and compost biochars (BC400, BC450, BC500, BC550, BC600, BC650) produced at a heating rate of 20 °C∙min^−1^; (**f**) the subtraction spectra of biochars produced at a heating rate of 20 °C min^−1^. The line indicates zero absorbance change for each subtraction spectrum.

**Figure 3 molecules-30-03365-f003:**
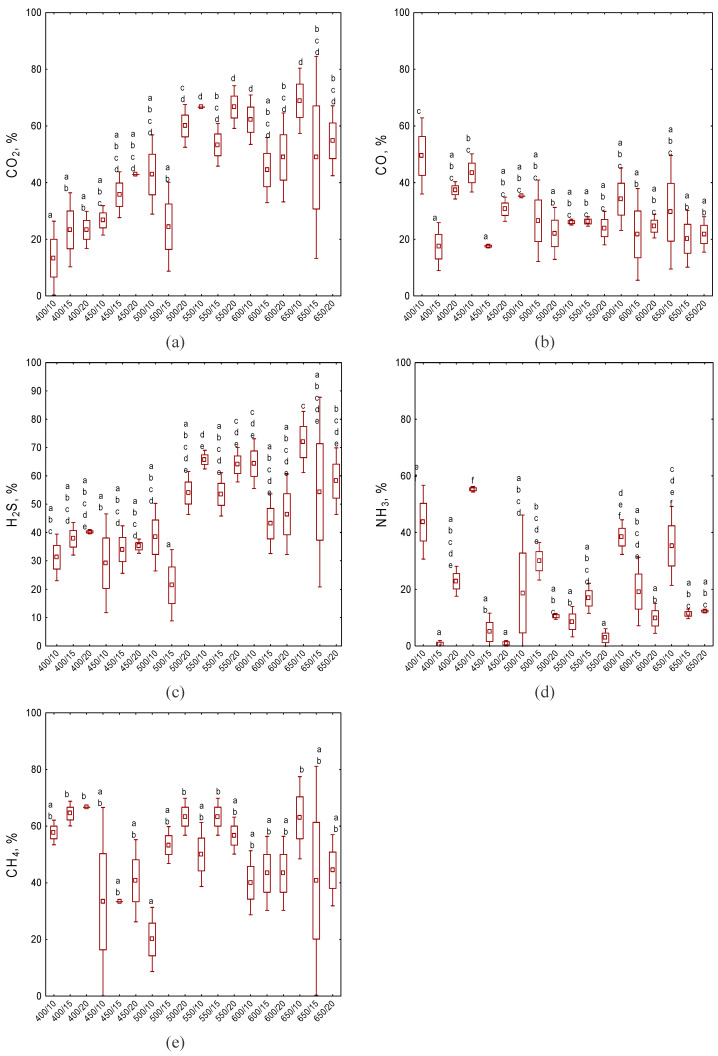
Summary of gas sorption efficiency of the analyzed biochars after 10 min of sorption tests, mean ± standard error: (**a**) CO_2_; (**b**) CO; (**c**) H_2_S; (**d**) NH_3_; (**e**) CH_4_; letters indicate affiliation to individual homogeneous groups determined based on Tukey’s test at a significance level of *p* < 0.05.

**Figure 4 molecules-30-03365-f004:**
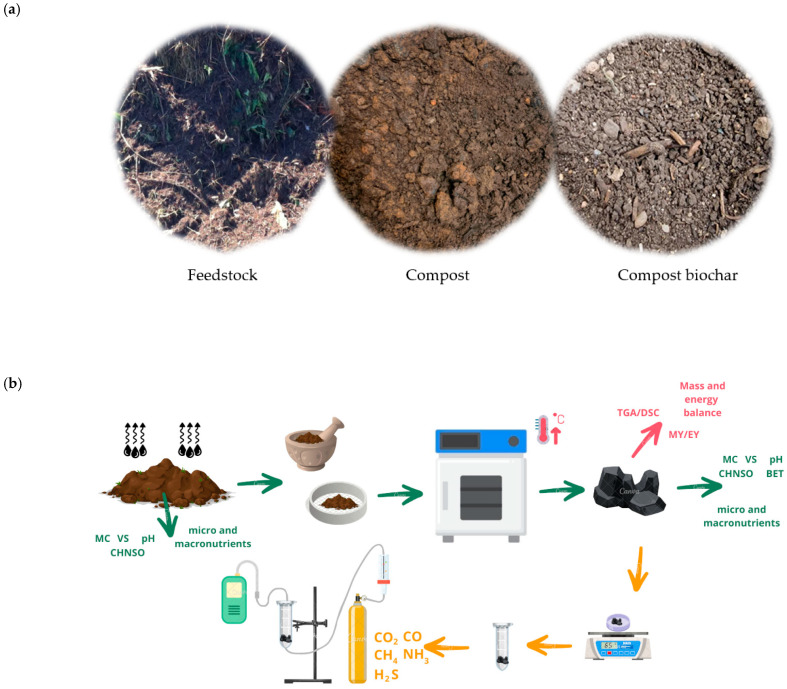
(**a**) Compost samples and compost biochar. (**b**) Experiment setup: configuration for the process of producing biochar from compost (green line), schematic diagram of adsorption test (yellow line), and mass and energy balance (red line).

**Table 1 molecules-30-03365-t001:** Compost and compost biochar properties (BC400, BC450, BC500, BC550, BC600, BC650) produced at a heating rate of 10 °C∙min^−1^. The heavy metal content of the compost biochars is shown in the Appendix A.

Variant	pH	Ash Content (AC)	Volatile Solids (VS)	Ca	K	Mg	Na	Total	P	C	H	N	S	O
	-	% d.m.		mg/kg d.m.					mg/kg	%				
400/15	8.4 ± 0.0	78.5 ± 0.1	21.5 ± 0.1	17,125 ± 145	6345 ± 45	2707 ± 10	1803 ± 3.0	27,980	5969 ± 1194	15.0 ± 3.0	1.0 ± 0.2	1.1 ± 0.2	0.3 ± 0.1	3.8 ± 0.8
400/20	8.6 ± 0.1	80.0 ± 0.0	20.0 ± 0.0	19,530 ± 120	6605 ± 35	3077 ± 5	1898 ± 1.0	31,110	5220 ± 1044	11.0 ± 2.0	0.5 ± 0.1	0.8 ± 0.2	0.3 ± 0.1	7.4 ± 1.5
400/40	8.5 ± 0.1	82.0 ± 0.1	18.0 ± 0.1	16,890 ± 165	6860 ± 30	2947 ± 15	1958 ± 2.0	28,655	5659 ± 1132	15.0 ± 3.0	0.8 ± 0.2	1.0 ± 0.2	0.3 ± 0.1	2.7 ± 0.5
450/10	9.0 ± 0.0	84.4 ± 0.0	15.6 ± 0.0	21,420 ± 225	6315 ± 55	5092 ± 30	1873 ± 1.5	34,700	5747 ± 1150	12.0 ± 2.0	0.8 ± 0.2	1.0 ± 0.2	0.3 ± 0.1	2.1 ± 0.4
450/15	9.5 ± 0.0	86.3 ± 0.2	13.7 ± 0.2	17,765 ± 25	6830 ± 150	3002 ± 20	1873 ± 2.5	29,470	5839 ± 1168	9.8 ± 1.2	0.4 ± 0.1	0.7 ± 0.1	0.3 ± 0.1	2.4 ± 0.4
450/20	9.3 ± 0.0	81.4 ± 0.1	18.6 ± 0.1	18,355 ± 195	6615 ± 15	3337 ± 40	2148 ± 2.0	30,455	5678 ± 1136	15.0 ± 3.0	0.9 ± 0.2	1.0 ± 0.2	0.3 ± 0.1	1.6 ± 0.3
500/10	9.4 ± 0.0	85.3 ± 0.1	14.7 ± 0.1	17,410 ± 275	6365 ± 25	3367 ± 45	1868 ± 1.5	29,010	5598 ± 1120	13.0 ± 3.0	0.5 ± 0.1	0.8 ± 0.2	0.3 ± 0.1	0.4 ± 0.1
500/15	9.6 ± 0.0	88.0 ± 0.2	12.0 ± 0.2	16,120 ± 220	5985 ± 10	2877 ± 5	1733 ± 2.5	26,715	5197 ± 1040	10.0 ± 2.0	0.4 ± 0.1	0.7 ± 0.1	0.3 ± 0.1	0.9 ± 0.2
500/20	9.6 ± 0.1	85.4 ± 0.0	14.6 ± 0.0	16,390 ± 115	6360 ± 5	2877 ± 10	1808 ± 2.5	27,435	5776 ± 1155	10.0 ± 2.0	0.3 ± 0.1	0.6 ± 0.1	0.3 ± 0.1	3.4 ± 0.7
550/10	8.6 ± 0.0	89.0 ± 0.0	11.0 ± 0.0	19,795 ± 175	7865 ± 65	3417 ± 35	2263 ± 1.5	33,340	6865 ± 1373	8.3 ± 1.7	0.2 ± 0.0	0.5 ± 0.1	0.3 ± 0.1	1.7 ± 0.3
550/15	9.1 ± 0.0	90.8 ± 0.0	9.2 ± 0.0	20,235 ± 330	7890 ± 65	3577 ± 35	2263 ± 0.5	33,965	7246 ± 1449	8.8 ± 1.8	0.1 ± 0.0	0.5 ± 0.1	0.3 ± 0.1	3.3 ± 0.7
550/20	9.3 ± 0.1	88.8 ± 0.1	11.2 ± 0.1	18,060 ± 105	7095 ± 10	3177 ± 15	2053 ± 1.5	30,385	6557 ± 1312	10.0 ± 2.0	0.1 ± 0.0	0.7 ± 0.1	0.3 ± 0.1	1.0 ± 0.2
600/10	9.1 ± 0.0	88.1 ± 0.2	11.9 ± 0.2	18,460 ± 105	7665 ± 60	3382 ± 35	2168 ± 1.0	31,675	6953 ± 1391	9.3 ± 1.8	0.2 ± 0.1	0.6 ± 0.1	0.3 ± 0.1	0.8 ± 0.2
600/15	9.1 ± 0.0	86.6 ± 0.0	13.4 ± 0.0	18,915 ± 305	7405 ± 105	3217 ± 20	2013 ± 1.0	31,550	6266 ± 1253	8.5 ± 1.8	0.2 ± 0.0	0.5 ± 0.1	0.3 ± 0.1	2.3 ± 0.5
600/20	8.8 ± 0.1	87.7 ± 0.1	12.3 ± 0.1	19,120 ± 130	7005 ± 80	3157 ± 30	1963 ± 1.5	31,245	5843 ± 1169	11.0 ± 2.0	0.3 ± 0.1	0.6 ± 0.12	0.3 ± 0.1	1.5 ± 0.3
650/10	9.4 ± 0.0	90.1 ± 0.0	9.9 ± 0.0	18,165 ± 80	7785 ± 65	3242 ± 10	2148 ± 1.0	31,340	6749 ± 1350	9.2 ± 1.8	0.2 ± 0.1	0.4 ± 0.1	0.3 ± 0.1	0.1 ± 0.0
650/15	9.5 ± 0.1	90.7 ± 0.1	9.3 ± 0.1	15,975 ± 45	7390 ± 125	2967 ± 15	2143 ± 2.0	28,475	5991 ± 1198	10.0 ± 2.0	0.1 ± 0.0	0.4 ± 0.1	0.3 ± 0.1	0.4 ± 0.1
650/20	9.4 ± 0.1	89.5 ± 0.0	10.5 ± 0.0	18,050 ± 125	8005 ± 170	3257 ± 25	2228 ± 0.5	31,540	6355 ± 1271	9.1 ± 1.8	0.1 ± 0.0	0.4 ± 0.1	0.3 ± 0.1	1.3 ± 0.3
Compost	7.3 ± 0.1	75.2 ± 0.2	24.8 ± 0.2	17,760 ± 135	6155 ± 30	2697 ± 10	1793 ± 2.5	28,405	5835 ± 1167	13.0 ± 3.00	1.2 ± 0.2	1.1 ± 0.2	0.4 ± 0.1	8.7 ± 0.4

**Table 2 molecules-30-03365-t002:** Energy and mass balance of the pyrolysis process for individual process parameters.

Temperature °C	Heating Rates °C min^−1^	Substrate		External Energy	Biochar	Gas	
		Mass of Substrate Used to Produce 1 g of Biochar, g	Energy Contained in Raw Material Used to Produce 1 g of Biochar, J	External Energy Needed to Produce 1 g of Biochar, J	Energy Contained in 1 g of Biochar, J	Mass of Gas Generated During the Production of 1 g of Biochar, g	Energy Contained in Gas After Production of 1 g of Biochar, J
550	10	1.197	6567	287	5655	0.197	1199
600	10	1.207	6625	296	5731	0.207	1190
650	10	1.211	6645	411	4452	0.211	2604

## Data Availability

Data are contained within the article and Appendix A.

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
