# Peer review of "Applying Compost Biochar for Gas Adsorption—Effects of Pyrolysis Conditions"

_molecules, 2025, doi:10.3390/molecules30163365_

Round 1

Reviewer 1 Report

Comments and Suggestions for Authors

The manuscript explores the compost, as a feedstock for biochar production. Sylwia Stegenta-Dąbrowska, Marta Galik used mature compost as a substrate, biochar was prepared by controlling the pyrolysis temperature (400-650°C) and heating rate (10-20°C min-1) and revealed a significant correlation between pyrolysis temperature and the sorption characteristics of compost-derived biochars. This study demonstrates good novelty, but lacks discussion on whether different pyrolysis temperatures may lead to fertility decline when compost is pyrolyzed into biochar. Therefore, Minor revision has to be done before this manuscript could be accepted for publication in the Molecules.

Minor comments:

  1. 1. In Abstract part, the language requires reorganization and further refinement. Suggest restructuring to follow "Purpose-Methods-Results-Conclusions" framework.

  1. 2. In Keywords part, present the following keywords in alphabetical order: "biochar", "carbon monoxide", "compost utilization", "compost valorization", "gas adsorption" ,"gas purification" ,"greenhouse gas", "pyrolysis".

  1. 3. The manuscript exhibits inconsistent formatting, such as "CO2" should be "CO2", "H2S" should be "H2S", and "CH4" should be "CH4". A thorough check is required.

  1. The labels"A" and "B"' for subfigures in Figure 2 should be uniformly changed to lowercase letters "a" and "b" to maintain consistent formatting throughout the manuscript.

  1. Could you clarify whether the primary objective was to optimize pyrolysis conditions (temperature/heating rate) for compost-derived biochar's gas adsorption performance, or to evaluate its feasibility for specific applications (e.g., composting vs. biogas purification)? The abstract emphasizes both aims, but the focus could be sharpened.

  1. The manuscript reports superior CO2/CH4/H2S adsorption at 550–650°C (Lines 21–30). What mechanistic insights (e.g.,pore structure vs. functional groups) explain this divergence from CO/NH3 trends?

  1. Lines 60–63 cite Silva et al. (2013) for compost standards. Are there updated guidelines (e.g., EU/IBI) relevant to your feedstock?

  1. Table 1: Specifythe units for "AC" (ash content) and "VS" (volatile solids).

  1. Discuss the policy implications, such as the cost-benefit analysis for composting facilities.

Author Response

The authors sincerely thank the reviewers for their valuable comments and efforts. We have carefully considered and incorporated all suggested revisions.

The discussion section has been improved based on the suggestions provided by the other reviewers.

This changes have been made in lines: 67-73;80-83;87-114.

Comment 1: In Abstract part, the language requires reorganization and further refinement. Suggest restructuring to follow "Purpose-Methods-Results-Conclusions" framework.
Response 1: Thank you for your valuable feedback. I have revised the abstract in accordance with your suggestions and restructured it following the "Purpose–Methods–Results–Conclusions" framework. The language has also been refined to improve clarity and scientific tone. This changes have been made in lines: 14-40

Comment 2: In Keywords part, present the following keywords in alphabetical order: "biochar", "carbon monoxide", "compost utilization", "compost valorization", "gas adsorption" ,"gas purification" ,"greenhouse gas", "pyrolysis".
Response 2: Thank you for your comment. The keywords have been reordered alphabetically as requested:
biochar, carbon monoxide, compost utilization, compost valorization, gas adsorption, gas purification, greenhouse gas, pyrolysis. This changes have been made in lines: 41-44.

Comment 3: The manuscript exhibits inconsistent formatting, such as "CO2" should be "CO2", "H2S" should be "H2S", and "CH4" should be "CH4". A thorough check is required.
Response 3: Thank you for your thorough review and attention to detail. We sincerely apologize for the inconsistent formatting of chemical formulas throughout the manuscript. We have carefully reviewed the entire document and corrected all instances where chemical formulas were not properly formatted with subscripts. Specifically, we have ensured that:
"CO2" is consistently formatted as "CO₂"
"H2S" is consistently formatted as "H₂S"
"CH4" is consistently formatted as "CH₄"
All other chemical formulas throughout the manuscript have also been checked and corrected to maintain consistent formatting. We appreciate your meticulous review, which has helped improve the overall quality and professionalism of our manuscript.

Comment 4: The labels "A" and "B"' for subfigures in Figure 2 should be uniformly changed to lowercase letters "a" and "b" to maintain consistent formatting throughout the manuscript.
Response 4: The labels of figure 2 have been changes according to reviewers recommendation. 

Comment 5: Could you clarify whether the primary objective was to optimize pyrolysis conditions (temperature/heating rate) for compost-derived biochar's gas adsorption performance, or to evaluate its feasibility for specific applications (e.g., composting vs. biogas purification)? The abstract emphasizes both aims, but the focus could be sharpened.
Response 5: Thank you for this insightful comment regarding the clarity of our research objectives. The primary objective of this study was to evaluate the practical feasibility of compost-derived biochar for specific gas adsorption applications, particularly to determine its suitability for composting emission control versus biogas purification. The optimization of pyrolysis conditions (temperature and heating rate) served as a methodological approach to achieve this practical goal rather than being the primary research focus. In response to your suggestion, we have expanded the conclusions section to better emphasize the practical applications and industrial potential of this technology.

Comment 6: The manuscript reports superior CO2/CH4/H2S adsorption at 550–650°C (Lines 21–30). What mechanistic insights (e.g.,pore structure vs. functional groups) explain this divergence from CO/NH3 trends?
Response 6:  Thank you for this valuable comment regarding the mechanistic insights behind the divergent adsorption behavior. In response to your feedback, we have expanded the analysis in the discussion section, adding comprehensive information about the influence of mineral salts and functional groups on the different gas adsorption mechanisms observed in our study.

Comment 7: Lines 60–63 cite Silva et al. (2013) for compost standards. Are there updated guidelines (e.g., EU/IBI) relevant to your feedstock?
Response 7: Thank you for this comment regarding compost quality standards. Yes, there are updated guidelines relevant to our feedstock, and we actually cite the International Biochar Initiative (IBI) standards in our manuscript. 

Comment 8: Table 1: Specifythe units for "AC" (ash content) and "VS" (volatile solids).
Response 8:  Thank you for this comment regarding the clarity of units in Table 1. We have clarified the units for "AC" (ash content) and "VS" (volatile solids) in Table 1. 

Comment 9: Discuss the policy implications, such as the cost-benefit analysis for composting facilities.
Response 9: Thank you for this interesting suggestion regarding policy implications and cost-benefit analysis for composting facilities. This is indeed a fascinating topic for future consideration. However, we believe that the manuscript is already comprehensive in scope, and additional economic analysis would, in our opinion, divert the reader's attention from the main focus of our work, which is the technical evaluation of compost-derived biochar for gas adsorption applications. We appreciate your comment and consider this an excellent direction for future research.

Reviewer 2 Report

Comments and Suggestions for Authors

Dear authors

The study titled “The Applying Composts’ Biochar for Gas Adsorption - Effect of 2 Pyrolysis Conditions’’ presents a comprehensive study on the use of compost-derived biochar for gas adsorption, focusing on the effects of pyrolysis conditions. The research is well-structured and addresses an important environmental challenge by exploring the potential of biochar to mitigate emissions from composting processes; however, minor revisions are required.

1- The authors must clarify why compost, despite its lower organic content compared to other feedstocks, is a viable candidate for biochar production. Additionally, the gap in research on gas mixtures (as opposed to pure gases) is well-highlighted.

  1. The authors must justify selecting these specific parameters (e.g., why 400–650°C and not higher or lower).
  2. The authors must clarify why certain properties (e.g., carbon content) did not follow expected trends (e.g., no clear increase with temperature).

4- The inverse relationship between pyrolysis temperature and CO adsorption is particularly interesting. However, the discussion could delve deeper into the mechanisms behind these observations. For example:

Why do higher temperatures favor CO2 and H2S adsorption but not CO and NH3?

How do the functional groups (as indicated by FTIR) correlate with adsorption performance?

5- The authors should discuss the practical implications of these findings, such as the feasibility of scaling up the process for industrial applications.

6-Figure 1: The labels for the y-axes (e.g., "Volume pore size, cm³·g⁻¹") are not fully legible. Consider increasing the font size or adjusting the layout.

7- Figure 3 needs more improvement; the labels for the homogeneous groups (a, b, c, etc.) are small and could be enlarged for better readability.

8-Table 1 is comprehensive but could be simplified for clarity. For example, the heavy metal content could be moved to the supplementary material, with only key elements (Ca, K, Mg, etc.) retained in the main table.

Best regards

Author Response

Thank you very much for your thoughtful and constructive review of our manuscript titled “The Applying Composts’ Biochar for Gas Adsorption – Effect of Pyrolysis Conditions.”
We truly appreciate your positive assessment of the study's structure and relevance, as well as your recognition of its contribution to addressing environmental challenges.

We have carefully considered your suggestions and made the necessary minor revisions as requested.

Comment 1: The authors must clarify why compost, despite its lower organic content compared to other feedstocks, is a viable candidate for biochar production. Additionally, the gap in research on gas mixtures (as opposed to pure gases) is well-highlighted.
Response 1: Thank you for your valuable comment. We agree with your observation regarding the need to clarify the suitability of compost as a feedstock for biochar production. Accordingly, we have revised the manuscript to include a justification, highlighting the advantages of compost despite its lower organic content. Also we have expanded the introduction to better emphasize the research gap concerning multicomponent gas systems. This changes have been made in lines: 67-73;80-83;87-114.

Comment 2: The authors must justify selecting these specific parameters (e.g., why 400–650°C and not higher or lower).
Response 2: Thank you for your valuable comment regarding the pyrolysis conditions. In response, we have revised the relevant paragraph to clearly and concisely summarize the optimal parameters for pyrolysis of biodegradable waste, based on both our previous findings (Syguła et al., 2019) and literature data (Sethupathi et al., 2017). This changes have been made in lines: 537-542.

Comment 3: The authors must clarify why certain properties (e.g., carbon content) did not follow expected trends (e.g., no clear increase with temperature).
Response 3: Thank you for this important observation. We have expanded the discussion to clarify why certain properties, particularly carbon content, did not follow the expected trends with increasing pyrolysis temperature.

Comment 4:

The inverse relationship between pyrolysis temperature and CO adsorption is particularly interesting. However, the discussion could delve deeper into the mechanisms behind these observations. For example:
-Why do higher temperatures favor CO2 and H2S adsorption but not CO and NH3?
-How do the functional groups (as indicated by FTIR) correlate with adsorption performance?
Response 4: Thank you for this insightful comment and for drawing attention to the need for further investigation of the mechanisms underlying the observed gas adsorption behavior. We have expanded the discussion in Section 2.4 (Sorption Mechanism) to address both questions raised.

Comment 5: The authors should discuss the practical implications of these findings, such as the feasibility of scaling up the process for industrial applications.
Response 5: Thank you for your feedback. We appreciate your insights. Our observations concerning the practical implications of these findings are detailed in the conclusions chapter. This changes have been made in lines: 636-641

Comment 6: Figure 1: The labels for the y-axes (e.g., "Volume pore size, cm³·g⁻¹") are not fully legible. Consider increasing the font size or adjusting the layout.
Response 6: We appreciate your attention to this matter. We have made an adjustment by increasing the font size by two points for improved readability in figure 1. Thank you.

Comment 7: Figure 3 needs more improvement; the labels for the homogeneous groups (a, b, c, etc.) are small and could be enlarged for better readability.
Response 7: Thank you for your feedback. I appreciate your input regarding the font size in the figure 3. However, I would like to note that increasing the font size could compromise legibility, as the letters may overlap. The drawing itself is of high quality, and by enlarging the overall image, all symbols will remain clearly readable. Thank you for your understanding.

Comment 8: Table 1 is comprehensive but could be simplified for clarity. For example, the heavy metal content could be moved to the supplementary material, with only key elements (Ca, K, Mg, etc.) retained in the main table.
Response 8: Thank you for your attention. Heavy metals have already been incorporated into the supplementary materials. Given the significance of this data, especially in relation to writing and discussing future publications in this field, we would prefer to have the data presented in a tabulated format as it appears. We appreciate your understanding.

Reviewer 3 Report

Comments and Suggestions for Authors

This manuscript presents a well-executed and experimentally rich investigation into the gas adsorption potential of biochars derived from compost under varying pyrolysis temperatures and heating rates. The study is timely and relevant, especially as the valorization of low-grade compost into functional adsorbent materials aligns with current circular economy and climate mitigation strategies. The inclusion of a wide gas matrix (CO2, CH4, H2S, CO, and NH3) and thorough statistical analyses enhances the scope and technical credibility of the study. Furthermore, the authors support their findings with FTIR, BET, and energy balance analysis, offering a multifaceted view of the structure-performance relationship.

  1. The manuscript uses both "specific surface area" and its abbreviation "SSA" interchangeably. For clarity and professional presentation, the authors should adopt a consistent terminology throughout the text and figures.
  2. While the novelty of using mature compost is acknowledged, the Introduction would benefit from a brief comparison with other biochar feedstocks, especially in terms of components. Additionally, framing the study with a focus on practical applications, such as emission control at composting sites versus biogas upgrading, would enhance its relevance.
  3. The selected pyrolysis temperature range (400-650 ℃) and heating rates (10–20 ℃/min) are presented without justification. The authors should elaborate on why these ranges were chosen, and it is highly recommended to provide the evidence from previous studies. Furthermore, the use of CO2 as the inert atmosphere during pyrolysis warrants explanation, given its potential to influence surface chemistry differently from N2 or air.
  4. Although compost-based biochar is less commonly studied for gas purification, prior works have explored similar biochar in emission control. The authors should clearly articulate the novel contribution of their study, whether it's the multi-gas testing setup, comparative heating rates, energy balance insights, or something else.
  5. The manuscript assesses adsorption for a gas mixture simulating composting emissions, but it omits any analysis of selectivity, particularly CO2 over N2 or CH4. Even a brief discussion or comparison with literature values would help position the compost biochar within the broader context of gas purification technologies. On the other hand, the figure 5 is missed.
  6. The CO2 adsorption enhancement is attributed to improved SSA and microporosity at higher temperatures, yet the highest SSA reported (~39.2 m2/g) is modest compared to conventional activated carbons. While the trend is valid, the authors might temper claims of "high performance" and instead emphasize material sustainability and cost-effectiveness.

Author Response

Dear Reviewer,

Thank you very much for this kind words. Authors really appreciate your valuable feedback. 

Comment 1: The manuscript uses both "specific surface area" and its abbreviation "SSA" interchangeably. For clarity and professional presentation, the authors should adopt a consistent terminology throughout the text and figures.
Response 1: 

Thank you for pointing out this inconsistency in our terminology. We acknowledge that using both "specific surface area" and "SSA" interchangeably throughout the manuscript reduces clarity of presentation.

We have carefully reviewed the entire manuscript and standardized the terminology. We have chosen to use "specific surface area (SSA)" upon first mention in each major section, and then consistently use "SSA" in subsequent references throughout the text and figures.

All figures, including Figure 1 caption and axis labels, have been updated to reflect this standardized terminology. We appreciate your attention to detail, which has helped improve the overall quality and consistency of our manuscript.

Comment 2: While the novelty of using mature compost is acknowledged, the Introduction would benefit from a brief comparison with other biochar feedstocks, especially in terms of components. Additionally, framing the study with a focus on practical applications, such as emission control at composting sites versus biogas upgrading, would enhance its relevance.
Response 2: Thank you for your valuable feedback. We have incorporated a concise comparison of the biochar derived from compost with other materials in the introduction. However, we have opted not to include information regarding the impact on biogas upgrading to maintain clarity and avoid potential confusion. We believe that a succinct and clear introduction enhances accessibility for readers. Detailed discussions on these topics are available in the results section.

Comment 3: The selected pyrolysis temperature range (400-650 ℃) and heating rates (10–20 ℃/min) are presented without justification. The authors should elaborate on why these ranges were chosen, and it is highly recommended to provide the evidence from previous studies. Furthermore, the use of CO2 as the inert atmosphere during pyrolysis warrants explanation, given its potential to influence surface chemistry differently from N2 or air.
Response 3: Thank you for your valuable comment regarding the pyrolysis conditions. In response, we have revised the relevant paragraph to clearly and concisely summarize the optimal parameters for pyrolysis of biodegradable waste, based on both our previous findings (Syguła et al., 2019) and literature data (Sethupathi et al., 2017). Also the used of CO2 as an inert gas for pyrolysis was explained. This changes have been made in lines: 537-542

Comment 4: Although compost-based biochar is less commonly studied for gas purification, prior works have explored similar biochar in emission control. The authors should clearly articulate the novel contribution of their study, whether it's the multi-gas testing setup, comparative heating rates, energy balance insights, or something else.
Response 4: Thank you for this valuable comment. We agree that it is important to clearly articulate the novel contributions of our study. We have revised the introduction to better highlight the specific novelty of our research.
Additionally, our study addresses the significant research gap identified in the literature regarding multicomponent gas systems under realistic conditions, moving beyond the pure gas or binary mixture studies that dominate current research but do not reflect practical applications in waste treatment facilities. This changes have been made in lines: 67-73;80-83;87-114; 144-150; 154-173.

Comment 5: The manuscript assesses adsorption for a gas mixture simulating composting emissions, but it omits any analysis of selectivity, particularly CO2 over N2 or CH4. Even a brief discussion or comparison with literature values would help position the compost biochar within the broader context of gas purification technologies. On the other hand, the figure 5 is missed.
Response 5:  With regard to the selectivity analysis, we are aware that dedicated selectivity measurements for specific gas pairs (CO₂/N₂, CO₂/CH₄) would provide valuable information on the positioning of our material in gas purification technology. However, it should be noted that the experimental variables used in our study, including the specific gas components and their concentrations in the mixture, introduces additional complexity to an already complex study. Our experimental design focused primarily on evaluating the adsorption performance of multiple gases under realistic composting emission conditions, rather than determining selectivity factors for individual gas pairs. Furthermore, our results show that different biochar variants exhibited optimal performance for different target gases (e.g., BC400/10 for CO, BC650/10 for CO₂ and H₂S), which suggests that selectivity may vary significantly depending on the specific application requirements. We appreciate this suggestion, and we will consider conducting dedicated CO₂/N₂ and CO₂/CH₄ selectivity experiments in our subsequent studies.

With regard to Figure 5. We apologize for the confusion in the text. There was a misunderstanding – we were referring to Figure 4b, which shows the experimental setup. The reference has been corrected in the manuscript.

Comment 6: The CO2 adsorption enhancement is attributed to improved SSA and microporosity at higher temperatures, yet the highest SSA reported (~39.2 m2/g) is modest compared to conventional activated carbons. While the trend is valid, the authors might temper claims of "high performance" and instead emphasize material sustainability and cost-effectiveness.
Response 6: Thank you for this important comment regarding the characteristics of biochar. We have revised the manuscript to tone down any statements that could be interpreted as “high performance” in an absolute sense. Instead, we now emphasize the advantages of composted biochar in terms of sustainability and cost-effectiveness. The discussion has also been revised.